environmental chemistry/environmental engineering/
chemical engineering

red mud, deNO$_x$, honeycomb catalyst,
selective catalytic reduction

**Authors for correspondence:**
Changming Li
e-mail: cmli@ipe.ac.cn
Jian Yu
e-mail: yujian@ipe.ac.cn

This article has been edited by the Royal Society of Chemistry, including the commissioning, peer review process and editorial aspects up to the point of acceptance.

# The utilization of red mud waste as industrial honeycomb catalyst for selective catalytic reduction of NO

Lin Huangfu[1,2], Abdullahi Abubakar[1,2], Changming Li[1], Yunjia Li[1,2], Chao Wang[3], Jian Yu[1] and Shiqiu Gao[1]

[1]State Key Laboratory of Multi-phase Complex Systems, Institute of Process Engineering, Chinese Academy of Sciences, Beijing 100190, People's Republic of China
[2]School of Chemistry and Chemical Engineering, University of Chinese Academy of Sciences, Beijing 100049, People's Republic of China
[3]School of Chemical Engineering, Xiangtan University, Xiangtan, Hunan 411105, People's Republic of China

JY, 0000-0003-0392-2556

As a new way for the high-value utilization of red mud (RM) waste, we proposed an improved approach to prepare the RM-based sludge/powder via the sulfuric acid hydrothermal dissolution and NH$_3$ aqueous precipitation route and then the RM-based industrial-sized honeycomb ($150 \times 150 \times 600$ mm) was successfully produced by the extrusion moulding method in pilot scale. The synthesized RM-based powdery/honeycomb catalyst exhibited more than 80% deNO$_x$ activity and good durability of H$_2$O and SO$_2$ above 350°C. But the decline of NO conversion was also observed above 350°C, which was confirmed to result from the increased oxygenation of NH$_3$ at high temperature. To improve the NO conversion at high temperature, NH$_3$ was shunted and injected into the catalyst bed at two different places (entrance and centre) to facilitate its uniform distribution, which relieved the oxidation of NH$_3$ and increased deNO$_x$ efficiency with 98% NO conversion at 400°C. This work explored the industrial application feasibility for the RM-based honeycomb catalyst as well as the possible solution to decrease the oxygenation of NH$_3$ at high temperature, which presented a valuable reference for the further pilot tests of RM catalyst in industry.

# 1. Introduction

Nitrogen oxides ($NO_x$), which contribute to photochemical smog, acid rain, ozone depletion and the greenhouse effect, are the main atmospheric pollutants. Selective catalytic reduction [1,2] with $NH_3$ (SCR-$NH_3$) is the most promising technology for eliminating $NO_x$ emission in industry, in which catalysts play the key role. Due to the brilliant SCR activity and $N_2$ selectivity at 300–400°C, commercial $V_2O_5$-$WO_3(MoO_3)/TiO_2$ [3–6] catalysts have been widely used for decades. Nevertheless, the V-based catalysts may also encounter some inevitable disadvantages, such as the high cost as well as the toxicity of vanadium pentoxide [7]. Therefore, the development of other eco-friendly and cost-effective SCR catalysts is still the research hotspot in the field.

As the possible substitution, the low cost and non-toxicity Fe-based catalysts [8–12] have attracted much attention in recent years. Red mud (RM) is a kind of solid waste produced during the production of alumina from bauxite, which contains the metal oxides including Fe, Al, Ti, Si, Na etc. Considering its particular composition, RM may be used as Fe-based SCR catalyst for $NO_x$ removal. But its deNO$_x$ efficiency is still unsatisfactory due to its high alkalinity and low specific surface [13,14]. The previously reported deNO$_x$ efficiency of RM catalyst only achieved 31% and 40% for CO-SCR [15] and $NH_3$-SCR [16], respectively. To increase the deNO$_x$ efficiency, our previous study [17] reported the preparation of RM-based powder catalyst with more than 80% NO conversion at high temperature above 350°C by a nitric acid-ball milling and neutralization-washing method. And its SCR performance was further increased after the activation of RM catalyst by $SO_2$, which displayed its great application potential in industry. However, the strong volatility of $HNO_3$ results in the low reaction efficiency between $HNO_3$ and RM, which may be not convenient for the industrial implementation to produce the RM-based honeycomb catalyst, and the easy oxidation of $NH_3$ at the temperature above 350°C may also prevent the further improvement of deNO$_x$ efficiency in industry.

In this work, we further proposed an improved approach to prepare the RM-based sludge/powder via the sulfuric acid hydrothermal dissolution and $NH_3$ aqueous precipitation route, and the RM-based industrial-sized honeycomb ($150 \times 150 \times 600$ mm) was successfully produced by the extrusion moulding method in pilot scale. Moreover, the process of injecting $NH_3$ in stages was proposed to decrease the oxidation of $NH_3$ and increase the NO conversion at the temperature above 350°C. The results reported here move forward the utilization of RM waste as industrial honeycomb catalyst, and facilitate its further pilot test of RM-based catalyst in the real flue gas.

# 2. Experimental procedure

## 2.1. Catalyst preparations

The preparation process of RM powdery/honeycomb catalyst is illustrated in figure 1. Original RM (table 1A) and 50 wt% $H_2SO_4$ were mixed at a molar ratio of 1/1.2. Then the whole mixture was transferred into steel autoclave and maintained at 150°C for 10 h, during which the alkaline/alkaline-earth metal was leached, the bulk oxides such as $Al_2O_3$, $Fe_2O_3$ and $TiO_2$ may also be partly dissolved by $H_2SO_4$ to form soluble aluminium sulfate, ferric sulfate and titanyl sulfate, but the $SiO_2$ was nearly insoluble [16]. After the treatment by $H_2SO_4$, the composite was washed several times and neutralized to pH value of 8 with aqueous $NH_3$ to remove the alkaline/alkaline-earth metal. The product of $Al(OH)_3$, $Fe(OH)_3$, $Ti(OH)_2$ and $SiO_2$ was obtained in the neutralization process. Then it was centrifuged and washed to produce the RM-based pug. The pug was either dried at 110°C and then calcined at 500°C for 3 h to prepare the powdery catalyst or was used directly to prepare the industrial-sized honeycomb catalyst ($150 \times 150 \times 600$ mm) by the extrusion moulding method. Finally, the calcined powdery sample was crushed and sieved to 30–50 mesh for SCR activity test in a fixed-bed quartz reactor (powder). The honeycomb was cut into 250 mm length with size of $30 \times 30$ mm.

## 2.2. Catalytic performance tests

The active test of powdery catalyst was conducted in a fixed-bed quartz reactor at different space velocity with 650 ppm NO, 650 ppm $NH_3$, 3 vol% $O_2$, 600 ppm $SO_2$ (when used), 10 vol% $H_2O$ (when used) and balanced with $N_2$. The activity measurement of honeycomb catalyst was carried out in a deNO$_x$ square-shape steel reactor. The simulated flue gas consisted of 3 vol% $O_2$, 600 ppm $SO_2$ (when used), 10 vol% $H_2O$ (when used) and balanced with $N_2$ at the ratio of $NH_3/NO = 1$. The experiment was carried out in a space velocity (SV) of 6000 h$^{-1}$ and 30 000 h$^{-1}$, respectively. The flue gas was continually

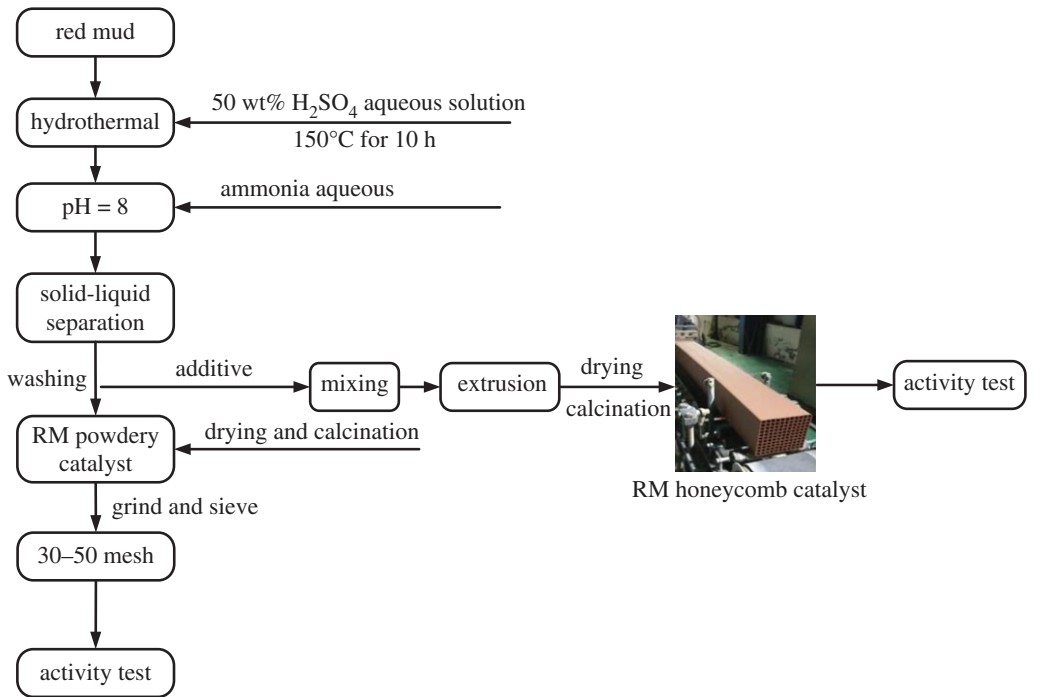

**Figure 1.** Schematic diagram for the preparation of RM catalyst/honeycomb.

**Table 1.** The composition information of powdery catalysts based on XRF results (wt%): (A) original RM, (B) fresh RM catalyst, (C) the sample used for 10 h, (D) the sample used for 20 h, (E) the sample used for 30 h, (F) the sample used for 40 h. Reaction condition: [NO] = [$NH_3$] = 650 ppm, [$O_2$] = 3 vol%, [$SO_2$] = 600 ppm, [$H_2O$] = 10 vol%, $N_2$ balanced.

| sample (%) | A | B | C | D | E | F |
|---|---|---|---|---|---|---|
| $Fe_2O_3$ | 44.18 | 45.13 | 44.50 | 44.96 | 45.52 | 45.34 |
| $Al_2O_3$ | 19.92 | 15.42 | 15.52 | 15.61 | 15.97 | 15.47 |
| $TiO_2$ | 8.01 | 8.25 | 8.1 | 8.15 | 7.67 | 7.98 |
| $SiO_2$ | 13.43 | 23.69 | 23.37 | 23.16 | 22.79 | 22.81 |
| $SO_3$ | 0.64 | 6.14 | 6.81 | 6.33 | 6.56 | 7.02 |
| $Na_2O$ | 10.97 | 0.24 | 0.23 | 0.29 | 0.3 | 0.24 |
| $CaO$ | 1.29 | 0.03 | 0.05 | 0.02 | 0.04 | 0.03 |
| $K_2O$ | 0.03 | 0.02 | 0.04 | 0.02 | 0.02 | 0.00 |

monitored by an ABB online gas analyser (ABB AO2020, Germany). Moreover, Gasmet portable FT-IR analyser (Gasmet DX4000, Finland) was used for testing the oxidation of $NH_3$ at different SV.

In order to decrease the oxidation of $NH_3$ and increase the NO conversion at the temperature above 350°C, the process of injecting $NH_3$ in stages over RM powdery catalyst was performed. As shown in figure 2, a new powdery catalyst (30–50 mesh) test device was designed with two fixed-bed quartz reactors to inject $NH_3$ at different places of catalytic bed. One part of $NH_3$ was injected into the first fixed-bed quartz reactor and the other $NH_3$ was directly injected into the second fixed-bed. The total mass of RM catalyst in the two fixed-bed quartz reactors is the same as the original test in the single-bed reactor, and the other reaction conditions are also identical.

The NO conversion, $N_2O$ yield and $NH_3$ conversion were calculated according to the following equations:

$$\text{NO conversion} = \frac{[\text{NO}]_{in} - [\text{NO}]_{out}}{[\text{NO}]_{in}} \times 100\%, \tag{2.1}$$

$$N_2O \text{ yield} = [N_2O]_{out} - [N_2O]_{in} \tag{2.2}$$

and
$$\text{NH}_3 \text{ conversion} = \frac{[\text{NH}_3]_{in} - [\text{NH}_3]_{out}}{[\text{NH}_3]_{in}} \times 100\%. \tag{2.3}$$

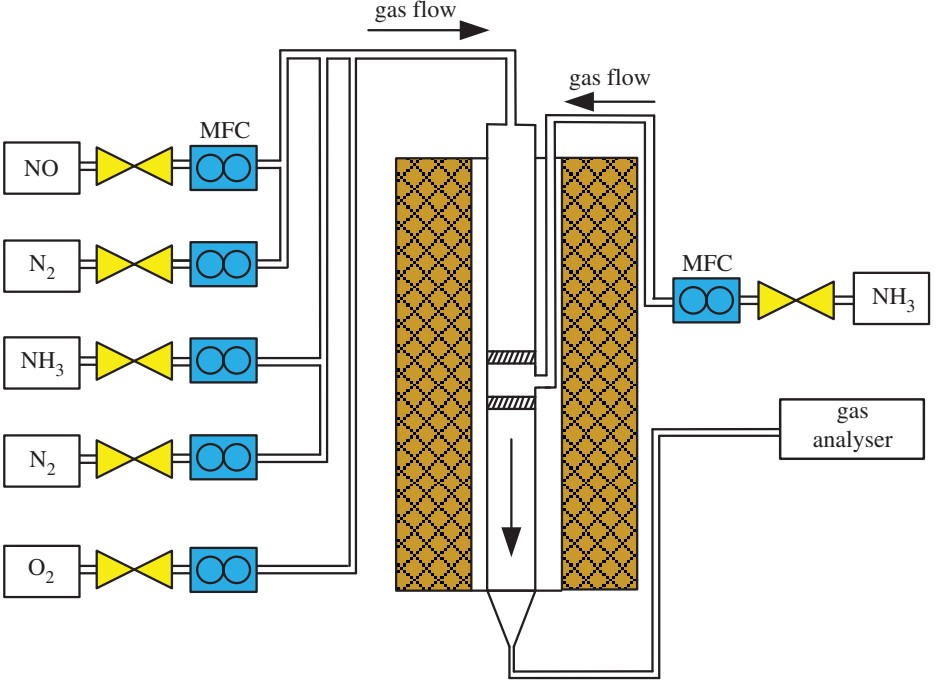

**Figure 2.** Schematic diagram of the staged injecting $NH_3$ process. MFC, mass flow control.

Where $[NO]_{in}$ and $[NO]_{out}$ represent the concentration of gaseous NO in the inlet and outlet, respectively.

## 2.3. Catalytic characterization

The X-ray fluorescence (XRF) spectrometer (AXIOS-MAX, PANalytical BV, Holland) was used to determine the chemical composition of catalysts. X-ray diffraction (XRD) patterns were recorded on an X-ray diffractometer (Empyrean, PANalytical BV, The Netherlands) in the $2\theta$ range of 5–90° with a step size of $0.4372°\,s^{-1}$ operating at 40 kV and 40 mA using Cu Kα radiation. The thermal analysis experiments were carried out in air atmosphere on a thermogravimetry/differential thermal analysis (TG/DTA 6300 produced by NSK). The mass of sample and the flow rate of air were 5 mg and $150\,ml\,min^{-1}$, respectively. The heating rate was $10°C\,min^{-1}$ from 30°C to 1000°C. A nitrogen adsorption–desorption apparatus (ASAP 2020, Micromeritics Instrument Corp., USA) was used to determine the surface area and pore-size distribution of samples at 77 K. The mass of samples was 0.1 g, and each of them was degassed at 200°C for 10 h before they were measured. The morphology and microstructure of samples were recorded on a SU8020 scanning electron microscope (SEM, Hitachi, Japan). The internal microstructures of samples were observed by a JEM-2010 transmission electron microscopy (TEM, JEOL, Japan) at an accelerating voltage of 200 kV.

# 3. Results and discussion

## 3.1. Structure and morphology of powdery catalysts

Table 1 summarizes the main composition of original RM, fresh RM catalyst and used RM samples (RM catalysts after SCR reaction). Due to the complexity of RM, the trace amounts of Zr, P, Ga, Cl, Ni, etc. are lower than 0.1% and not listed. The original RM (table 1A) mainly consists of $Fe_2O_3$ (44.18%), which is the active component for SCR reaction. $Al_2O_3$, $TiO_2$ and $SiO_2$ are common support components in catalytic system. Generally, the high $Na_2O$ content and other alkaline-earth metals in the original RM will interact with the major active components during catalytic reaction, resulting in the decrease in surface area and activity [13]. After the acid-hydrothermal and neutralization method, the majority of alkali/alkaline-earth metals were removed, and the fresh RM catalyst (table 1B) was obtained with the active $Fe_2O_3$ supported on $SiO_2$–$Al_2O_3$–$TiO_2$ composite. Figure 3a shows the typical XRD patterns of the synthesized RM catalysts. It can be seen that the main crystalline phases of fresh RM catalyst are α-$Fe_2O_3$, $TiO_2$ and $SiO_2$. The peak at

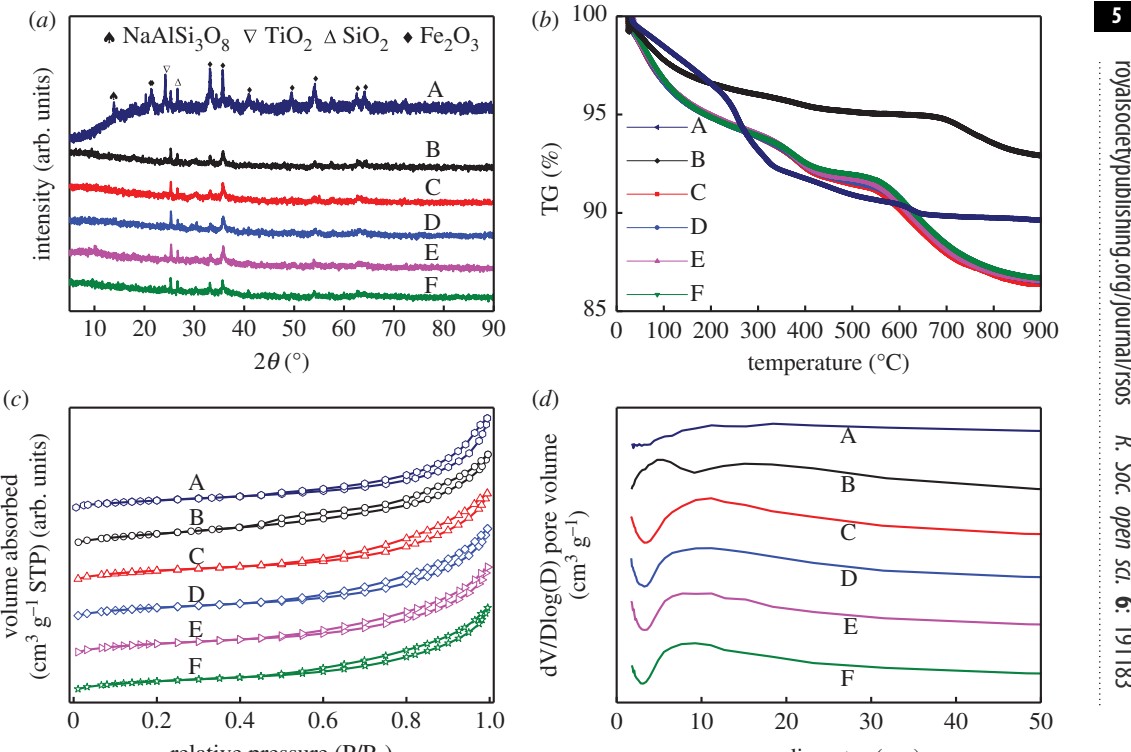

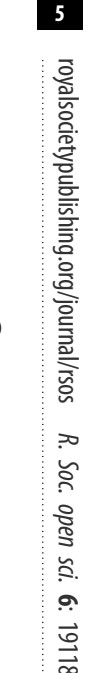

**Figure 3.** The XRD (*a*), TG (*b*), $N_2$ adsorption–desorption isotherms curves (*c*) together with the corresponding pore-size distributions (*d*) patterns of: (A) original RM, (B) fresh RM catalyst, (C) the sample used for 10 h, (D) the sample used for 20 h, (E) the sample used for 30 h and (F) the sample used for 40 h. Reaction condition: [NO] = [NH₃] = 650 ppm, [O₂] = 3 vol%, [SO₂] = 600 ppm, [H₂O] = 10%, N₂ balanced.

13.89° for albite disappears when compared with the original RM, which is consistent with the results of XRF. After the dissolution–precipitation process, the peak intensity $Fe_2O_3$ weakened, indicating the decreased particle size and increased dispersion of active Fe species of the obtained RM catalyst. The thermogravimetric (TG) analysis of samples is illustrated in figure 3*b*. It is generally accepted that the weight loss below 300°C belongs to the loss of adsorbed water. The second region (300–500°C) of weight loss can be attributed to the dehydroxylation for original and fresh RM samples. Besides, there is an obvious weight loss of fresh RM catalyst between 650 and 800°C, which can be assigned to the decomposition of residual sulfates.

The BET surface area measurement is also performed to have a deep insight into the change of surface area and pore-size distribution. Figure 3 illustrates the $N_2$ adsorption–desorption isotherms curves (figure 3*c*) and the corresponding pore-size distribution (figure 3*d*) of catalysts, whose isotherms are close to type IV with a $H_3$-type hysteresis loop, indicating the typical mesoporous characteristics. Obviously, the pore-size distribution of fresh RM catalyst (figure 3*c*-B) is broader than original RM (figure 3*c*-A). The BET surface area, pore volumes and average pore diameter are listed in table 2. It is found that the surface area of fresh RM catalyst increases significantly when compared with the original RM. The large specific surface area of RM catalyst and its small pore diameter may facilitate the dispersion and exposure of the active sites.

As shown in figure 3*a*, there is no obvious crystal transition/generation before and after the SCR reaction (A–E) over RM catalysts in the presence of $H_2O$ and $SO_2$. Except for the content of $SO_4^{2-}$, little difference in compositions is observed between the fresh RM catalyst and catalysts after reaction according to table 1. The weight loss of the used samples (figure 3*b*) below 300°C is owing to the loss of water. The decomposition of ammonium sulfate or ammonium bisulfate [18,19] is also observed between 300 and 400°C from the residual sulfate in the used RM samples. And the weight loss at the temperature range of 400–800°C is due to the decomposition of metal sulfates. In addition, no obvious change of the BET results (figure 3*c,d*) for these used samples signifies the structure stability of our RM catalyst.

To have a better understanding of the morphology structure of RM catalyst/honeycomb, the SEM observation was carried out. In the case of fresh RM catalyst, many small irregular grains can be

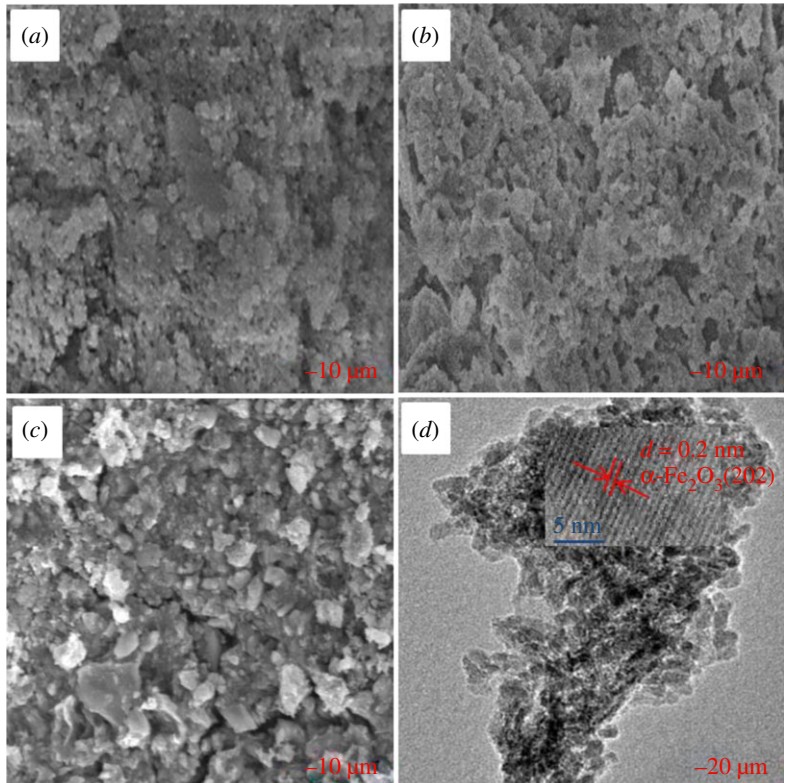

**Figure 4.** SEM images of fresh RM catalyst (*a*), RM catalyst used for 40 h (*b*) and RM honeycomb catalyst (*c*) together with the TEM image of fresh RM catalyst (*d*). Reaction condition: [NO] = [NH$_3$] = 650 ppm, [O$_2$] = 3 vol%, [SO$_2$] = 600 ppm, [H$_2$O] = 10 vol%, N$_2$ balanced.

**Table 2.** BET surface area and pore structure results of the powdery catalysts: (A) original RM, (B) fresh RM catalyst, (C) the sample used for 10 h, (D) the sample used for 20 h, (E) the sample used for 30 h, (F) the sample used for 40 h. Reaction condition: [NO] = [NH$_3$] = 650 ppm, [O$_2$] = 3 vol%, [SO$_2$] = 600 ppm, [H$_2$O] = 10 vol%, N$_2$ balanced.

| catalyst | surface area (m$^2$ g$^{-1}$) | pore volume (cm$^3$ g$^{-1}$) | pore diameter (nm) |
|---|---|---|---|
| A | 83 | 0.210 | 10.1 |
| B | 103 | 0.211 | 8.2 |
| C | 98 | 0.207 | 8.5 |
| D | 97 | 0.210 | 8.7 |
| E | 101 | 0.207 | 8.2 |
| F | 97 | 0.199 | 8.1 |

observed with the size of micrometre grade (figure 4*a*), and there is no significant change in morphology after SCR reaction (figure 4*b*). Figure 4*c* shows that fine RM catalyst particles on the surface of honeycomb are well dispersed. Moreover, the TEM image of fresh RM catalyst in figure 4*d* identifies the lattice fringe phase of α-Fe$_2$O$_3$ (2 0 2) crystal phase with the lattice distance of 0.2 nm, which is consistent with the results of XRD in figure 3*a*.

In brief, the alkali/alkaline-earth metal elements of original RM can be eliminated by acid-hydrothermal dissolution and NH$_3$ aqueous neutralization method, and the as-prepared RM catalyst consists of active Fe$_2$O$_3$ component as well as the composite oxides of SiO$_2$–Al$_2$O$_3$–TiO$_2$. The acid and alkali treatment process also increases the specific surface area of RM catalyst, which facilitates to achieve better dispersion of the active Fe$_2$O$_3$ species. Moreover, no obvious difference was found between the fresh and used RM samples, indicating the good stability of our RM catalyst.

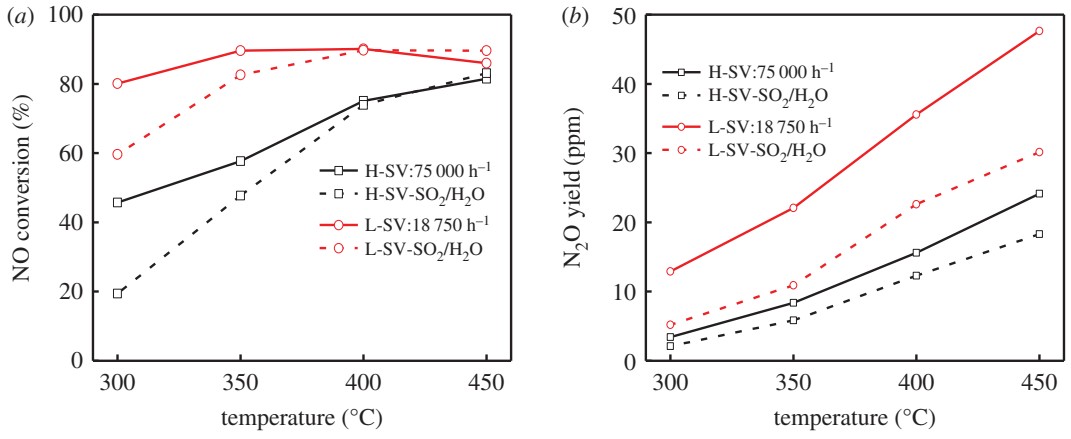

**Figure 5.** The NO conversion (*a*) and $N_2O$ yield (*b*) with/without $H_2O$ and $SO_2$ over powdery catalysts. Reaction condition: $[NO] = [NH_3] = 650$ ppm, $[O_2] = 3$ vol%, $[SO_2] = 600$ ppm, $[H_2O] = 10$ vol%, $N_2$ balanced.

## 3.2. Catalytic performance

### 3.2.1. The SCR activity of RM powdery catalyst

The NO conversion of RM powdery catalyst is investigated and illustrated in figure 5*a*. The temperature window for above 80% NO conversion ranged from 350 to 450°C (SV = 18 750 h$^{-1}$) in the absence of $H_2O$ and $SO_2$, which was much higher than that of 31 and 40% deNO$_x$ efficiency in the previous study [15,16]. The lower the SV, the longer the contact time of the gas and catalyst, resulting in the increase of NO conversion. But the produced $N_2O$ is also increased at low SV. The decrease in deNO$_x$ activity was observed for the addition of $H_2O$ and $SO_2$ at low temperature below 350°C. However, as the temperature increased, the inhibition effect of $H_2O$ and $SO_2$ decreased. Particularly, $H_2O$ and $SO_2$ slightly promoted the NO conversion of the RM catalyst above 400°C. Meanwhile, it can be found in figure 5*b* that the $N_2O$ formation decreased obviously in the presence of $H_2O$ and $SO_2$, especially at the low SV and high temperature. Generally, $H_2O$ and $SO_2$ may damage the low-temperature SCR activity, apparently due to the blocking of the active sites by ammonium sulfate [20] and the competitive adsorption between $H_2O$ and $NH_3$ [21,22]. For this reason, the $N_2O$ formation also decreased. As the temperature increased, the $H_2O$ could be easily desorbed from inactive Fe sites, resulting in the recovery of Fe active sites as well as the NO conversion. Besides, $H_2O$ could inhibit the oxidation of $NH_3$ at high temperature, which is of benefit to the improvement of SCR activity above 400°C. The observed effect of $H_2O$ and $SO_2$ on the deNO$_x$ activity of RM catalysts is consistent with these previous reports [17], and the RM catalyst may also have good stability and resistance to $H_2O$ and $SO_2$ above 350°C.

### 3.2.2. SCR performance of RM honeycomb catalyst

To explore the application feasibility of RM catalyst in industry, the RM honeycomb catalyst was further produced by extrusion moulding method. As shown in figure 6*a*, the NO conversion increased along with the increasing temperature at high SV (30 000 h$^{-1}$), but the best NO conversion was only 60%. To further increase the deNO$_x$ efficiency, the experiment at low SV was conducted. The RM honeycomb catalyst achieved more than 80% NO conversion in the temperature range of 350–450°C at 6000 h$^{-1}$, and the maximum NO conversion reached 87% at 400°C. With the temperature further increased, the NO conversion declined while the formation of $N_2O$ (figure 6*b*) increased dramatically. Additionally, on the basis of the fact that the exhaust usually contains water vapour and a certain amount of $SO_2$, the effects of $H_2O$ and $SO_2$ on the catalytic performance of RM honeycomb catalysts were further investigated. Although the addition of 10% $H_2O$ and 600 ppm $SO_2$ significantly suppressed the NO conversion at low temperature, the adverse effect progressively weakened with increased temperature. Meanwhile, the $N_2O$ formation was lower in the presence of $H_2O$ and $SO_2$, and best NO conversion could reach above 80% at 450°C with the $N_2O$ lower than 20 ppm on the condition. The SCR performance of our RM honeycomb catalyst is in good agreement of the results of RM powdery catalyst, demonstrating that it is feasible to use RM-based catalyst for denitrification in industrial applications.

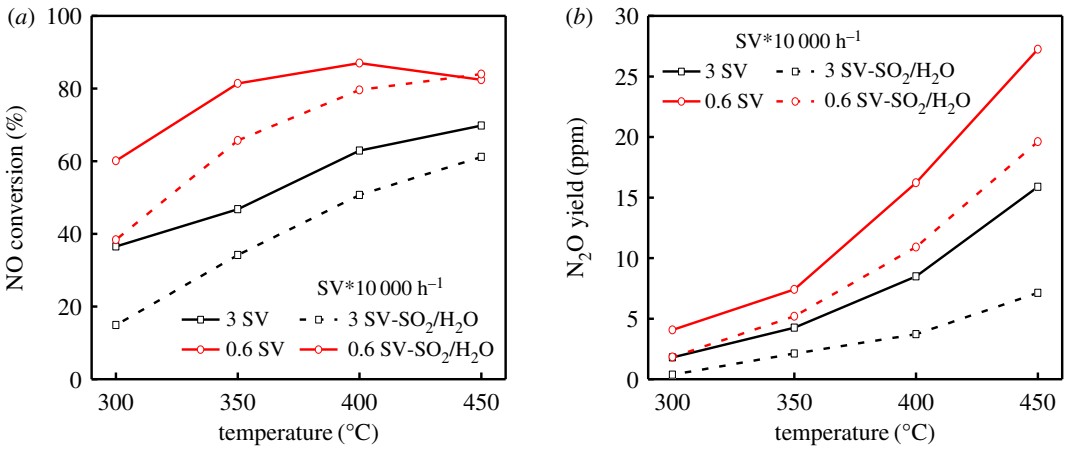

**Figure 6.** The NO conversion (*a*) and N$_2$O yield (*b*) with/without H$_2$O and SO$_2$ over RM honeycomb catalyst. Reaction condition: [NO] = [NH$_3$] = 650 ppm, [O$_2$] = 3 vol%, [SO$_2$] = 600 ppm, [H$_2$O] = 10 vol%, N$_2$ balanced, velocity = 3 N m s$^{-1}$.

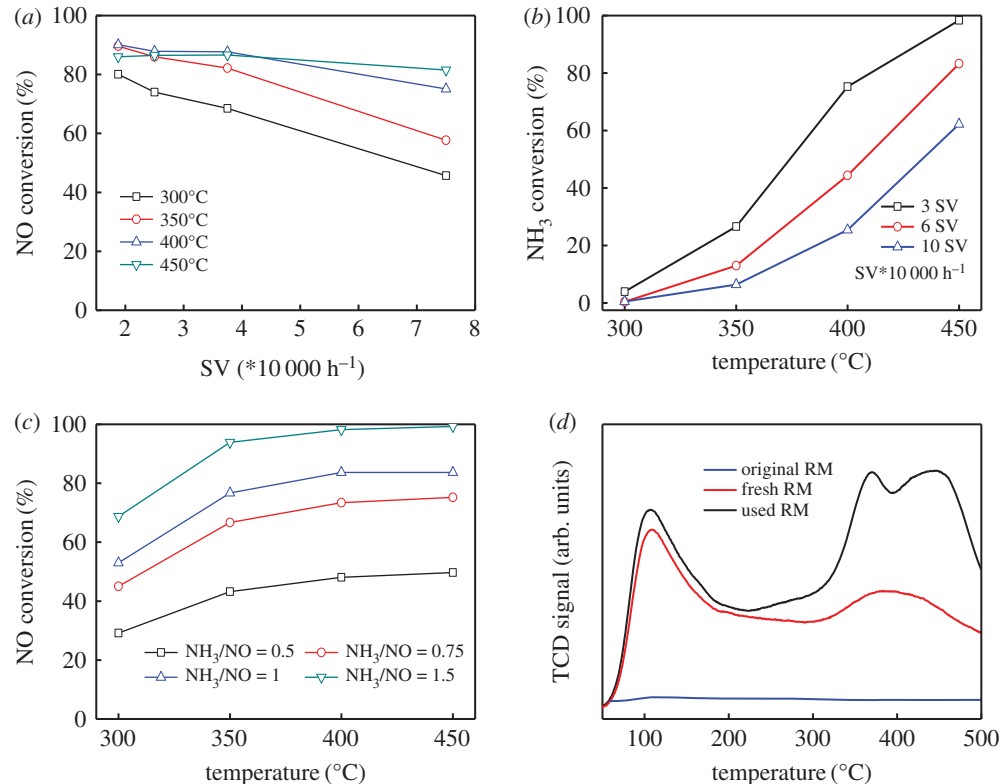

**Figure 7.** The effect of space velocity (*a*), NH$_3$ oxidation (*b*) and NH$_3$/NO ratio (*c*) together with the result of NH$_3$-TPD (*d*) over RM catalyst. Reaction condition: [NH$_3$] = 650 ppm, [NO] = 325–975 ppm, [O$_2$] = 3 vol%, N$_2$ balanced.

## 3.3. The effect of ammoxidation on SCR efficiency at the temperature above 350°C

### 3.3.1. The features of ammoxidation during the SCR reaction over RM catalysts

As stated above, it can be concluded that for both the powder and honeycomb RM catalyst, the deNO$_x$ efficiency is superior at low SV than that at high SV. Nevertheless, more N$_2$O was formed at low SV with low SCR selectivity, which prevented the further improvement of deNO$_x$ efficiency close to 100% at high temperature above 350°C. Therefore, more tests were done to reveal the features of ammoxidation during the SCR reaction. Figure 7*a* illustrates the effect of SV on SCR performance over powdery RM catalyst. It displayed that deNO$_x$ efficiency increased linearly and apparently at 300–350°C with the decrease in SV. But at high reaction temperature above 350°C, the decrease in SV had no obvious improvement on

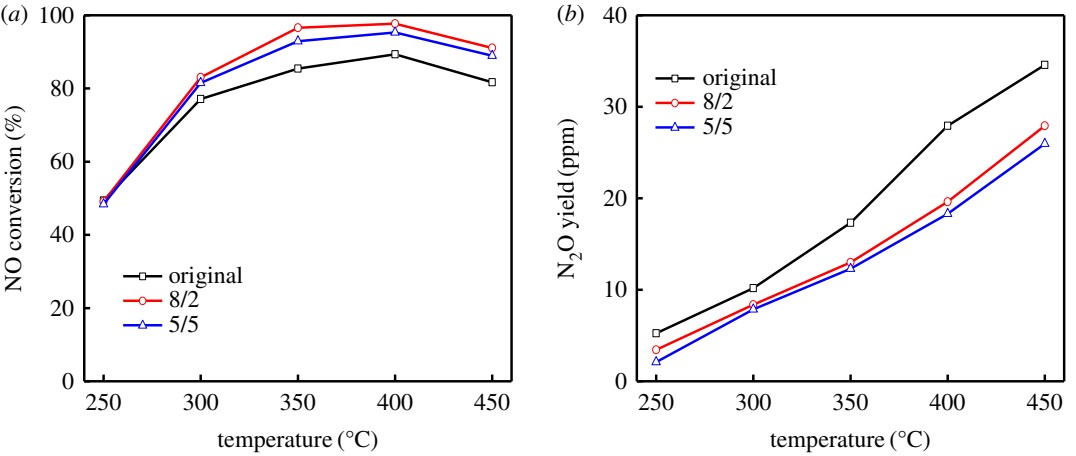

**Figure 8.** Comparison of NO conversion (*a*) and $N_2O$ yield (*b*) between the original single-bed process and the process of injecting $NH_3$ in stages. Reaction condition: $[NH_3] = 650$ ppm, $[NO] = 650$ ppm, $[O_2] = 3$ vol%, $N_2$ balanced, $SV = 30\,000\ h^{-1}$.

deNO$_x$ efficiency. The NO conversion was still lower than 90% even at low SV and high temperature, which was different from V-based catalyst with almost 100% deNO$_x$ efficiency on proper condition. Most interestingly, the NO conversion of 450°C even decreased at lower SV, implying that it is impossible to improve the high-temperature SCR activity by simply increasing the amount of catalyst.

The $NH_3$ oxidation measurement at different SV is observed in figure 7*b*. It exhibited that the $NH_3$ oxidation drastically increased above 350°C. As the SV decreased, the oxidation of $NH_3$ enhanced. Especially at 450°C, $NH_3$ was almost completely oxidized at the SV of $30\,000\ h^{-1}$ due to the longer contact time between catalyst and $NH_3$. Although NO conversion has been improved with the decrease in SV, the oxidation of $NH_3$ at low SV was much higher than that at high SV in the high-temperature range. The oxidation of $NH_3$ decreased the supply of $NH_3$ for the SCR reaction and thus prevented the achieved high NO conversion at low SV and high temperature for RM catalysts.

To further confirm this point, we investigated the effect of $NH_3$/NO ratio and the result is depicted in figure 7*c*. It can be seen that NO conversion increased with the increase in $NH_3$/NO ratio. And it reached the theoretical value at 400°C when the concentration of $NH_3$ was lower than NO. However, only 80% NO conversion was obtained when $NH_3$/NO was 1 and it achieved almost 100% NO conversion at the ratio of $NH_3$/NO = 1.5. Thus, it can be deduced that when $NH_3$ was sufficient such as the $NH_3$/NO ≥ 1, a large amount of $NH_3$ was oxidized and consumed at high temperature. As a contrast, the low ratio of $NH_3$/NO showed the low oxygenation efficiency of $NH_3$ with its high utilization efficiency. The results were also consistent with previous reports about the competition reaction between $NH_3$-SCR and $NH_3$-SCO ($NH_3$-selective catalytic oxidation) [8,23,24]. The -$NH_2$ intermediate species [25,26] activated by $NH_3$ preferentially reacted with NO at the lower $NH_3$ concentration. With the increase in $NH_3$/NO ratio, the oxidation of $NH_3$ was enhanced, resulting in the lack of $NH_3$ to react with NO. Accordingly, the maximum theoretical value of NO conversion cannot be reached owing to the severe oxidation of $NH_3$ at high-temperature range.

The oxidizability of $NH_3$ is related to the acid sites of RM catalysts, thus $NH_3$-TPD (temperature programmed desorption) test is carried out as profiled in figure 7*d*. It can be seen that the adsorption capacity of $NH_3$ increased significantly for the fresh RM sample compared with the original one due to the elimination of alkali/alkaline-earth metals. The desorption peak centred at about 100°C can be assigned to physically adsorbed $NH_3$. Another obvious desorption peak can be observed from 300 to 500°C, which may be due to the desorption of ionic $NH_4^+$ bound to strong Brønsted acid sites and coordinated $NH_3$ bound to Lewis acid sites. It was reported [17,27] that $NH_3$ was mainly absorbed on the Brönsted acid sites at relatively low temperature to form $NH_4^+$ and then react with NO to generate $N_2$ and $H_2O$, while the coordinated $NH_3$ bound to Lewis acid sites usually response for the SCR activity at high temperature. Both of the Brönsted acid sites and Lewis acid sites are important in the SCR/SCO reaction with $NH_3$. As for the used RM catalyst, the residual sulfate may further increase the $NH_3$ adsorption. These strong acid sites between 300 and 500°C will facilitate the strong adsorption of ammonia and will be also beneficial for the $NH_3$-SCO reaction, resulting in the decreased SCR selectivity. All of these results indicate that the severe oxidation of $NH_3$ resulted in the low actual ratio of $NH_3$/NO and thus low reaction efficiency of SCR.

### 3.3.2. The improved deNO$_x$ efficiency by injecting NH$_3$ in stages

In order to decrease the oxidation of NH$_3$ and increase the NO conversion at high temperature, the process of injecting NH$_3$ in stages over RM powdery catalyst was proposed. The effects of NH$_3$ concentration and distribution on the SCR reaction are illustrated in figure 8. Despite that the NO conversion also decreased with increasing temperature above 400°C, the new process demonstrated much more advantages in NO conversion than the original single-bed reactor (the black line). The NO conversion increased from 89 to 95% at 400°C after the introduction of injecting NH$_3$ in stages at 5/5 NH$_3$ distribution of the two parts. Particularly, the best NO conversion achieved 98% when the NH$_3$ distribution of the two parts is 8/2, which is also coincident with the result in figure 7c. Meanwhile, it is worth noting that the N$_2$O formation of our new process was much lower in the entire temperature range, which indicated the decrease in NH$_3$ oxidation with the improved N$_2$ selectivity. These results demonstrated that the injection of NH$_3$ in stages can significantly weaken the oxidation of NH$_3$ at high temperature, increase the utilization of NH$_3$ and improve the high-temperature deNO$_x$ efficiency. According to our experimental results, the NH$_3$ distribution of 8/2 for the two parts may be a suitable choice, and the process of injecting NH$_3$ in stages may facilitate to reach high deNO$_x$ efficiency at high temperature by reducing oxidation of NH$_3$ for the further application of RM catalysts in industry.

# 4. Conclusion

The RM-based powdery/honeycomb catalyst was successfully produced through the improved H$_2$SO$_4$-NH$_3$ pretreatment and subsequent extrusion process with above 80% deNO$_x$ activity and good durability of H$_2$O and SO$_2$ above 350°C. The characterization results of XRF, XRD and BET indicated that the elimination of alkali/alkaline-earth metal elements from original RM increased the specific surface area and facilitated to achieve better dispersion of the active components of our RM-based catalyst. Although NO conversion increased with the decrease in SV, the theoretical 100% NO conversion still cannot be obtained even at low SV. The further test of SV, NH$_3$/NO ratio, oxygenation efficiency of NH$_3$ as well as the NH$_3$-TPD confirmed that the severe oxidation of NH$_3$ over the strong acid sites of RM catalyst resulted in the low actual ratio of NH$_3$/NO at high temperature, and the lack of NH$_3$ prevented the achievement of ideal SCR reaction efficiency in theory. As an effective solution, the process of injecting NH$_3$ in stages was carried out and could significantly reduce the oxidation of NH$_3$ owing to the more reasonable distribution of NH$_3$ in the SCR reactor. The best NO conversion increased to 98% when the distribution of NH$_3$ is 8/2 in the two parts. The successful production of RM-based industrial-sized honeycomb catalyst as well as the improved SCR reaction efficiency by the process of injecting NH$_3$ in stages signify the application feasibility for the utilization of RM waste as industrial honeycomb deNO$_x$ catalyst.

Data accessibility. The data that support the findings of this study are openly available from electronic supplementary material, which includes the raw data of the experiments and characterizations.

Authors' contributions. J.Y., C.L. and S.G conceived and designed the study; L.H. performed the experiments and drafted the article. L.H., A.A., C.L., Y.L. and C.W. coordinated the study and revised the article. All authors gave final approval for publication.

Competing interests. We declare we have no competing interest.

Funding. This work was supported by International Science and Technology Cooperation Program of China (grant no. 2016YFE0128300), Natural Science Foundation of China (grant nos 21601192 and 21878310) and the open subject from State Key Laboratory of Multi-phase Complex Systems (grant no. MPCS-2019-0-03).

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
