## [Reviewer comments · Royal Society Open Science]

Review History

RSOS-191183.R0 (Original submission)

Review form: Reviewer 1

Is the manuscript scientifically sound in its present form?

Yes

Are the interpretations and conclusions justified by the results?

Yes

Is the language acceptable?

Yes

Do you have any ethical concerns with this paper?

No

Have you any concerns about statistical analyses in this paper?

No

Recommendation?

Major revision is needed (please make suggestions in comments)

Comments to the Author(s)

Dear Authors

Your work is devoted to the application red mud waster for the synthesis of catalysts for reduction of NO. The main attention was focused on the improvement of catalytic properties of such type catalytic systems. Unfortunately, some moments are unclear.

Keywords

In my opinion should be revised, for example, Selective Catalytic Reduction (SCR); NH₃-SCR; Red Mud catalyst; Honeycomb deNO_x Catalyst; Effect of H₂O and SO₂

Introduction

You wrote that the improving catalytic property is based on the sulfuric acid hydrothermal dissolution of red mud and then NH₃ aqueous precipitation. I suggest to discuss chemistry of this approach.

Experimental

- I think that Table 1 (Chemical composition of red mud and catalysts) should be moved to this section.

Table 1. Is this weight percent?

Results and Discussion

- Can chemical composition of red mud affect the properties of catalyst?

- According to Table 1, acid modification of red mud with H₂SO₄ leads to the leaching of Al and Sodium ions and increasing amount of Si and S. How can these changes affect the nature of active sites and therefore, affect the catalytic properties?

- Did you found TiOSO₄ salt in XRD?

- Figures 5 and 7. Please, revise X-axis.

- I suggest to compare efficiency of your catalysts with that of systems reported in literature

Review form: Reviewer 2

Is the manuscript scientifically sound in its present form?

Yes

Are the interpretations and conclusions justified by the results?

No

Is the language acceptable?

Yes

Do you have any ethical concerns with this paper?

No

Have you any concerns about statistical analyses in this paper?

No

Recommendation?

Major revision is needed (please make suggestions in comments)

Comments to the Author(s)

This work aims at the evaluation of the performance in the selective catalytic reduction of NO of a catalyst prepared from red mud waste. It is a very interesting approach since this catalytic process is still attracting the interest of many research groups and the valorization of industrial wastes is a priority in order to minimize the negative environmental impact of many industrial chemical processes. However, there are several points requiring to be addressed in order to clarify the information provided in this manuscript:

1. (Introduction, p. 4., l. 45) What does it mean that the low reaction efficiency between HNO₃ and RM may be not convenient for the industrial amplification (implementation)?
2. It would be interesting to include a picture of the RM honeycomb catalyst prepared in the present study.
3. (Experimental, p. 8, 19) They refer to the determination of BET and BJH, but this is meaningless, because these are two methods for the determination of specific surface area (BET) and pore size distribution (BJH), respectively.
4. How do the authors explain that, by comparing the chemical composition of original RM and fresh RM catalyst (after the acid-hydrothermal and neutralization steps), the weight (is it true? or is it molar?) percentages of Fe₂O₃ and TiO₂ remain almost unchanged, Al₂O₃ decreases and SiO₂ increases)?
5. (p. 10, l. 11) The BET is not a test, as previously indicated, and this method is usually used for the calculation of specific surface area (BET surface area), excepting for microporous materials. This method does not provide information about the pore size distribution, as is indicated.
6. It would be very useful to study catalysts by X-ray photoelectron spectroscopy (XPS) in order to get insights into the surface composition of fresh and used RM catalysts.
7. The scale of the x-axis of Figure 3D does not allow to distinguish clearly the evolution of pore size distribution curves. The representation until 100 nm, or less, could be enough.
8. Have they detected the formation of a particular metal sulfate, since the percentage of SO₃ is high. By considering the formation enthalpy of different potential metal sulfates, which is the most probable to be formed in the presence of Fe(III), Al(III) and Ti(IV)?
9. What is the reason of the slight increase in NO conversion in the presence of H₂O and SO₂ at 450°C (Figure 5A)?
10. The total acidity can be evaluated from NH₃-TPD data. An explanation, taking into account the chemical composition of original RM and fresh RM catalyst, must be given to justify the drastic difference between the corresponding desorption curves (Figure 7D). The existence of strong Brønsted acid sites and Lewis acid sites is put forward by the authors, but nothing is said about the nature of chemical species responsible of these acid sites.
11. It is necessary to include in situ DRIFTS experiments of reactants in order to elucidate the mechanisms involved in SCR activity and ammoxidation. In this sense, is it actually an ammoxidation process leading to nitriles by reaction of ammonia and oxygen, using alkenes as substrate?. If this is not the case, the nature of oxidation products must be explained.
12. A paragraph comparing the present results with data already reported in the literature should be very useful for readers.
13. Minor errors: (Figure 1) Molding additive?, Grind and sieve instead of Grind and seive; (p. 7, l. 22) N₂O formation, not information.

Therefore, it is an interesting methodological approach for the design of a new catalyst for NH₃-SCR, but these issues must be clarified before recommending the manuscript for publication.

Decision letter (RSOS-191183.R0)

12-Aug-2019

Dear Dr Yu:

Title: The utilization of red mud waste as industrial honeycomb catalyst for selective catalytic reduction of NO

Manuscript ID: RSOS-191183

The editor assigned to your manuscript has now received comments from reviewers. We would like you to revise your paper in accordance with the referee and Subject Editor suggestions which can be found below (not including confidential reports to the Editor). Please note this decision does not guarantee eventual acceptance.

Please submit your revised paper before 04-Sep-2019. Please note that the revision deadline will expire at 00.00am on this date. If we do not hear from you within this time then it will be assumed that the paper has been withdrawn. In exceptional circumstances, extensions may be possible if agreed with the Editorial Office in advance. We do not allow multiple rounds of revision so we urge you to make every effort to fully address all of the comments at this stage. If deemed necessary by the Editors, your manuscript will be sent back to one or more of the original reviewers for assessment. If the original reviewers are not available we may invite new reviewers.

RSC Associate Editor:
Comments to the Author:
(There are no comments.)

RSC Subject Editor:
Comments to the Author:
(There are no comments.)

Reviewers' Comments to Author:
Reviewer: 1

Comments to the Author(s)
Dear Authors

Your work is devoted to the application red mud waster for the synthesis of catalysts for reduction of NO. The main attention was focused on the improvement of catalytic properties of such type catalytic systems. Unfortunately, some moments are unclear.

Keywords

In my opinion should be revised, for example, Selective Catalytic Reduction (SCR); NH₃-SCR; Red Mud catalyst; Honeycomb deNO_x Catalyst; Effect of H₂O and SO₂

Introduction

You wrote that the improving catalytic property is based on the sulfuric acid hydrothermal dissolution of red mud and then NH₃ aqueous precipitation. I suggest to discuss chemistry of this approach.

Experimental

- I think that Table 1 (Chemical composition of red mud and catalysts) should be moved to this section.

Table 1. Is this weight percent?

Results and Discussion

- Can chemical composition of red mud affect the properties of catalyst?

- According to Table 1, acid modification of red mud with H₂SO₄ leads to the leaching of Al and Sodium ions and increasing amount of Si and S. How can these changes affect the nature of active sites and therefore, affect the catalytic properties?

- Did you found TiOSO₄ salt in XRD?

- Figures 5 and 7. Please, revise X-axis.

- I suggest to compare efficiency of your catalysts with that of systems reported in literature

Reviewer: 2

Comments to the Author(s)

This work aims at the evaluation of the performance in the selective catalytic reduction of NO of a catalyst prepared from red mud waste. It is a very interesting approach since this catalytic process is still attracting the interest of many research groups and the valorization of industrial wastes is a priority in order to minimize the negative environmental impact of many industrial chemical processes. However, there are several points requiring to be addressed in order to clarify the information provided in this manuscript:

1. (Introduction, p. 4., l. 45) What does it mean that the low reaction efficiency between HNO₃ and RM may be not convenient for the industrial amplification (implementation)?
2. It would be interesting to include a picture of the RM honeycomb catalyst prepared in the present study.
3. (Experimental, p. 8, 19) They refer to the determination of BET and BJH, but this is meaningless, because these are two methods for the determination of specific surface area (BET) and pore size distribution (BJH), respectively.
4. How do the authors explain that, by comparing the chemical composition of original RM and fresh RM catalyst (after the acid-hydrothermal and neutralization steps), the weight (is it true? or is it molar?) percentages of Fe₂O₃ and TiO₂ remain almost unchanged, Al₂O₃ decreases and SiO₂ increases)?
5. (p. 10, l. 11) The BET is not a test, as previously indicated, and this method is usually used for the calculation of specific surface area (BET surface area), excepting for microporous materials. This method does not provide information about the pore size distribution, as is indicated.
6. It would be very useful to study catalysts by X-ray photoelectron spectroscopy (XPS) in order to get insights into the surface composition of fresh and used RM catalysts.
7. The scale of the x-axis of Figure 3D does not allow to distinguish clearly the evolution of pore size distribution curves. The representation until 100 nm, or less, could be enough.
8. Have they detected the formation of a particular metal sulfate, since the percentage of SO₃ is high. By considering the formation enthalpy of different potential metal sulfates, which is the most probable to be formed in the presence of Fe(III), Al(III) and Ti(IV)?
9. What is the reason of the slight increase in NO conversion in the presence of H₂O and SO₂ at 450°C (Figure 5A)?
10. The total acidity can be evaluated from NH₃-TPD data. An explanation, taking into account the chemical composition of original RM and fresh RM catalyst, must be given to justify the drastic difference between the corresponding desorption curves (Figure 7D). The existence of strong Brönsted acid sites and Lewis acid sites is put forward by the authors, but nothing is said about the nature of chemical species responsible of these acid sites.
11. It is necessary to include in situ DRIFTS experiments of reactants in order to elucidate the mechanisms involved in SCR activity and ammoxidation. In this sense, is it actually an ammoxidation process leading to nitriles by reaction of ammonia and oxygen, using alkenes as substrate?. If this is not the case, the nature of oxidation products must be explained.
12. A paragraph comparing the present results with data already reported in the literature should be very useful for readers.
13. Minor errors: (Figure 1) Molding additive?, Grind and sieve instead of Grind and seive; (p. 7, l. 22) N₂O formation, not information.

Therefore, it is an interesting methodological approach for the design of a new catalyst for NH₃-SCR, but these issues must be clarified before recommending the manuscript for publication.

Author's Response to Decision Letter for (RSOS-191183.R0)

See Appendix A.

RSOS-191183.R1 (Revision)

Review form: Reviewer 1

Is the manuscript scientifically sound in its present form?

Yes

Are the interpretations and conclusions justified by the results?

Yes

Is the language acceptable?

Yes

Do you have any ethical concerns with this paper?

No

Have you any concerns about statistical analyses in this paper?

No

Recommendation?

Accept as is

Comments to the Author(s)

-

Review form: Reviewer 2

Is the manuscript scientifically sound in its present form?

Yes

Are the interpretations and conclusions justified by the results?

Yes

Is the language acceptable?

Yes

Do you have any ethical concerns with this paper?

No

Have you any concerns about statistical analyses in this paper?

Yes

Recommendation?

Accept as is

Comments to the Author(s)

The authors have made an important effort to adequately answer most of questions raised in the revision process. This has allowed to improve and clarify the information provided in the manuscript, and I could recommend its publication.

Decision letter (RSOS-191183.R1)

04-Oct-2019

Dear Dr Yu:

Title: The utilization of red mud waste as industrial honeycomb catalyst for selective catalytic reduction of NO

Manuscript ID: RSOS-191183.R1

It is a pleasure to accept your manuscript in its current form for publication in Royal Society Open Science. The chemistry content of Royal Society Open Science is published in collaboration with the Royal Society of Chemistry.

RSC Associate Editor:
Comments to the Author:
Both reviewers recommend that the manuscript can now be accepted.

RSC Subject Editor:
Comments to the Author:
(There are no comments.)

Reviewer(s)' Comments to Author:
Reviewer: 1

Comments to the Author(s)

-

Reviewer: 2

Comments to the Author(s)

The authors have made an important effort to adequately answer most of questions raised in the revision process. This has allowed to improve and clarify the information provided in the manuscript, and I could recommend its publication.

Appendix A

Response to Reviewers

Reviewer: 1

Comments to the Author(s)

Dear Authors

Your work is devoted to the application red mud waster for the synthesis of catalysts for reduction of NO. The main attention was focused on the improvement of catalytic properties of such type catalytic systems. Unfortunately, some moments are unclear.

Keywords

Issue 1: In my opinion should be revised, for example, Selective Catalytic Reduction (SCR); NH₃-SCR; Red Mud catalyst; Honeycomb deNO_x Catalyst; Effect of H₂O and SO₂

Discussion: The keywords are revised as suggested in the revised manuscript.

p.2, Line 17: rephrase: Red mud; DeNO_x; Honeycomb catalyst; Selective catalytic reduction

Introduction

Issue 2: You wrote that the improving catalytic property is based on the sulfuric acid hydrothermal dissolution of red mud and then NH₃ aqueous precipitation. I suggest to discuss chemistry of this approach.

Discussion: This is a good suggestion for better understanding the preparation process of RM catalyst. As shown in Table 1, the main composition of original RM is Fe₂O₃, Al₂O₃, SiO₂, TiO₂, and alkaline/alkaline earth metal. The 50 wt.% H₂SO₄ was use to remove alkaline/alkaline earth metal through acid base neutralization. At the same time, the bulk oxides such as Fe₂O₃ and Al₂O₃ may also partly dissolved by H₂SO₄ to form soluble ferric sulfate and aluminum sulfate. After the treatment by H₂SO₄, ammonia aqueous was added to generate Fe(OH)₃ and Al(OH)₃ precipitate. Then the Fe(OH)₃ and Al(OH)₃ transformed to active Fe₂O₃ and inert Al₂O₃ after calcined at 500 °C for 3 h. The Fe₂O₃ can act as the main active component, and the other oxides (e.g., Al₂O₃, TiO₂, and SiO₂) will be as the support components in RM based catalyst.

Table 1 The composition information of powdery catalysts based on XRF results (wt.%): (a) original RM, (b) fresh RM catalyst, (c) the used sample for 10 h, (d) the used sample for 20 h, (e) the used sample for 30 h, (f) the used sample for 40 h. Reaction condition: [NO] = [NH₃] = 650 ppm, [O₂] = 3 vol.%, [SO₂] = 600 ppm, [H₂O] = 10 vol.%, N₂ balanced.

Sample	a	b	c	d	e	f
Fe ₂ O ₃ (%)	44.18	45.13	44.50	44.96	45.52	45.34
Al ₂ O ₃ (%)	19.92	15.42	15.52	15.61	15.97	15.47
TiO ₂ (%)	8.01	8.25	8.1	8.15	7.67	7.98
SiO ₂ (%)	13.43	23.69	23.37	23.16	22.79	22.81
SO ₃ (%)	0.64	6.14	6.81	6.33	6.56	7.02
Na ₂ O (%)	10.97	0.24	0.23	0.29	0.3	0.24
CaO (%)	1.29	0.03	0.05	0.02	0.04	0.03
K ₂ O (%)	0.03	0.02	0.04	0.02	0.02	0.00

p.4, Line 9: rephrase: The preparation process of RM powdery/honeycomb catalyst was illustrated in **Fig. 1**. Original RM (**Table 1a**) and 50 wt.% H₂SO₄ were mixed at a molar ratio of 1/1.2. Then the whole mixture was transferred into steel autoclave and maintained at 150 °C for 10 h, during which the alkaline/alkaline earth metal was leached, the bulk oxides such as Al₂O₃, Fe₂O₃ and TiO₂ may also partly dissolved by H₂SO₄ to form soluble aluminum sulfate, ferric sulfate and

titanyl sulfate, but the SiO₂ was nearly insoluble [16]. After the treatment by H₂SO₄, the composite was washed for several times and neutralized to pH value of 8 with NH₃ aqueous to remove the alkaline/alkaline earth metal. The product of Al(OH)₃, Fe(OH)₃, Ti(OH)₂ and SiO₂ was obtained in the neutralization process.

Reference

16. Sushil S, Batra VS. 2008 Catalytic applications of red mud, an aluminium industry waste: A review. *Appl. Catal. B-Environ.* **81**, 64-77. (10.1016/j.apcatb.2007.12.002)

Experimental

Issue 3:- I think that Table 1 (Chemical composition of red mud and catalysts) should be moved to this section.

Table 1. Is this weight percent?

Discussion: The composition information (**Table 1**) of powdery catalysts was obtained by the X-ray fluorescence (XRF) spectrometer, which displayed the weight percent of elements or oxides. Also, Table 1 is moved as suggested.

Results and Discussion

Issue 4:- Can chemical composition of red mud affect the properties of catalyst?

Discussion: As reported in the literature [16], original RM mainly consists of Fe₂O₃, Al₂O₃, SiO₂, TiO₂, Na₂O, CaO and K₂O etc. The calcium and sodium oxides can interact with the major active components, leading to crystallographic and morphologic changes with the decrease of surface area and activity. Therefore, the alkaline/alkaline earth metal was eliminated during the sulfuric acid hydrothermal dissolution in this work, and the generation of Fe species in the NH₃ aqueous precipitation facilitated to achieve better dispersion of the active components of our RM based catalyst.

Reference

16. Sushil S, Batra VS. 2008 Catalytic applications of red mud, an aluminium industry waste: A review. *Appl. Catal. B-Environ.* **81**, 64-77. (10.1016/j.apcatb.2007.12.002)

Issue 5:- According to Table 1, acid modification of red mud with H₂SO₄ leads to the leaching of Al and Sodium ions and increasing amount of Si and S. How can these changes affect the nature of active sites and therefore, affect the catalytic properties?

Discussion: In this work, the elimination of alkali/alkaline-earth metal elements from original RM increased the specific surface area and facilitated to achieve better dispersion of the active components of our RM based catalyst. It is consistent with the literature reports [13, 16], which demonstrated that dissolution–precipitation method can decrease the Ca and Na red mud content, and increase its specific surface, leading to higher activity and extended life periods for both hydrogenations and oxidations. As shown in **Table 1**. The RM catalyst in our study consisted of the active Fe₂O₃ species and SiO₂-Al₂O₃-TiO₂ composite support. The leached Al ion transformed to Al(OH)₃ precipitate with the addition of NH₃ aqueous and then the Al₂O₃ was obtained as catalytic support of the RM catalyst after calcined at 500 °C for 3 h. As for the increase of S (in the formation of SO₄²⁻), the catalytic activity of SCR reaction may be promoted by SO₄²⁻, which may increase the amount of acid sites as well as the absorption of NH₃, resulting in the improved catalytic activity of SCR reaction. Therefore, we revised the corresponding part for better

understanding the preparation process of RM catalyst.

p.4, Line 9: rephrase: The preparation process of RM powdery/honeycomb catalyst was illustrated in **Fig. 1**. Original RM (**Table 1a**) and 50 wt.% H_2SO_4 were mixed at a molar ratio of 1/1.2. Then the whole mixture was transferred into steel autoclave and maintained at 150 °C for 10 h, during which the alkaline/alkaline earth metal was leached, the bulk oxides such as Al_2O_3 , Fe_2O_3 and TiO_2 may also partly dissolved by H_2SO_4 to form soluble aluminum sulfate, ferric sulfate and titanyl sulfate, but the SiO_2 was nearly insoluble [16]. After the treatment by H_2SO_4 , the composite was washed for several times and neutralized to pH value of 8 with NH_3 aqueous to remove the alkaline/alkaline earth metal. The product of $\text{Al}(\text{OH})_3$, $\text{Fe}(\text{OH})_3$, $\text{Ti}(\text{OH})_2$ and SiO_2 was obtained in the neutralization process.

Reference

13. Ordonez S. 2008 Comments on "Catalytic applications of red mud, an aluminium industry waste: A review". *Appl. Catal. B-Environ.* **84**, 732-733. (10.1016/j.apcatb.2008.06.001)

16. Sushil S, Batra VS. 2008 Catalytic applications of red mud, an aluminium industry waste: A review. *Appl. Catal. B-Environ.* **81**, 64-77. (10.1016/j.apcatb.2007.12.002)

Issue 6:- Did you found TiOSO_4 salt in XRD?

Discussion: As suggested, we compared the XRD pattern with the standard card carefully, no TiOSO_4 salt was found in XRD.

Issue 7:- Figures 5 and 7. Please, revise X-axis.

Discussion: Figures 5 and 7 have been checked and corrected in the revised manuscript as suggested.

Fig. 5 The NO conversion (A) and N_2O yield (B) with/without H_2O and SO_2 over powdery catalysts. Reaction condition: $[\text{NO}] = [\text{NH}_3] = 650$ ppm, $[\text{O}_2] = 3$ vol.%, $[\text{SO}_2] = 600$ ppm, $[\text{H}_2\text{O}] = 10$ vol.%, N_2 balanced.

Fig. 7 The effect of space velocity (A), NH₃ oxidation (B) and NH₃/NO ratio (C) together with the result of NH₃-TPD (D) over RM catalyst. Reaction condition: [NH₃] = 650 ppm, [NO] = 325-975 ppm, [O₂] = 3 vol.%, N₂ balanced.

Issue 8:- I suggest to compare efficiency of your catalysts with that of systems reported in literature.

Discussion: This is a good suggestion. The utilization of red mud waste as SCR catalyst has been studied before. Mohapatro et al. [15] reported the red mud catalyst with 31% deNO_x efficiency at 400 °C for CO-SCR, and it increased to 92% when plasma was cascaded with red mud catalyst. The red mud SCR catalyst prepared by Bhattacharyya et al. [16] obtained 40% NO conversion. Although the deNO_x efficiency for red mud catalyst is still unsatisfactory, it demonstrates the feasibility to use red mud waste as high-value SCR catalyst. The main reason for the dissatisfied deNO_x performance is the poor dispersity of active Fe species and the adverse effect of alkalis in RM.

p.12, Line 14: rephrase: The NO conversion of RM powdery catalyst was investigated and illustrated in **Fig. 5A**. The temperature window for above 80% NO conversion ranged from 350 °C to 450 °C (SV = 18,750 h⁻¹) in the absence of H₂O and SO₂, which was much higher than that of 31% and 40% deNO_x efficiency in the previous study[15, 16].

Reference

- Mohapatro S, Rajanikanth BS. 2012 Dielectric barrier discharge cascaded with red mud waste to enhance NO_x removal from diesel engine exhaust. *IEEE Trns. Dielectr. Electr. Insul.* **19**, 641-647. (10.1109/tdei.2012.6180259)
- Sushil S, Batra VS. 2008 Catalytic applications of red mud, an aluminium industry waste: A

Reviewer: 2

Comments to the Author(s)

This work aims at the evaluation of the performance in the selective catalytic reduction of NO of a catalyst prepared from red mud waste. It is a very interesting approach since this catalytic process is still attracting the interest of many research groups and the valorization of industrial wastes is a priority in order to minimize the negative environmental impact of many industrial chemical processes. However, there are several points requiring to be addressed in order to clarify the information provided in this manuscript:

1. (Introduction, p. 4., l. 45) What does it mean that the low reaction efficiency between HNO₃ and RM may be not convenient for the industrial amplification (implementation?)?

Discussion: Due to the strong volatility and instability of HNO₃, only low concentration of HNO₃ solution (20 wt.%) can be used, which may decrease the reaction efficiency between HNO₃ and RM. Besides, the toxic volatile is harmful to the environment, limiting the production of large-scale RM deNO_x catalyst *via* the nitric acid-ball milling and neutralization-washing method.

p.3, Line 17: rephrase: However, the strong volatility of HNO₃ results in the low reaction efficiency between HNO₃ and RM, which may be not convenient for the industrial implementation to produce the RM based honeycomb catalyst, and the easy oxidation of NH₃ at the temperature above 350 °C may also prevent the further improvement of deNO_x efficiency in industry.

2. It would be interesting to include a picture of the RM honeycomb catalyst prepared in the present study.

Discussion: The picture of the RM honeycomb catalyst was included in **Fig. 1** as suggested.

Fig. 1 Schematic diagram for preparation of RM catalyst/honeycomb.

3. (Experimental, p. 8, 19) They refer to the determination of BET and BJH, but this is meaningless, because these are two methods for the determination of specific surface area (BET) and pore size distribution (BJH), respectively.

Discussion: The corresponding part has been revised as suggested.

p.8, Line 5: rephrase: A nitrogen adsorption-desorption apparatus (ASAP 2020, Micromeritics Instrument Corp, USA) was used to determine the surface area and pore size distribution of samples at 77 K.

4. How do the authors explain that, by comparing the chemical composition of original RM and fresh RM catalyst (after the acid-hydrothermal and neutralization steps), the weight (is it true? or is it molar?) percentages of Fe₂O₃ and TiO₂ remain almost unchanged, Al₂O₃ decreases and SiO₂ increases)?

Discussion: After acid-hydrothermal treatment, the alkaline/alkaline earth metal together with part Al, Fe and Ti oxides were dissolved, but the SiO₂ is nearly insoluble. The subsequent neutralization and washing process will remove the alkaline/alkaline earth metal element and recover of the Fe₂O₃, Al₂O₃ and TiO₂. However, the different solubility (Al>Fe>Ti>Si) [16] resulted in the different recover rate of these Al, Fe and Ti species during the neutralization process. More Al was lost during the process, but SiO₂ was almost completely retained. Ultimately, we observed the increased percentage of SiO₂, almost unchanged percentage of Fe₂O₃ and TiO₂, and decreased percentage of Al₂O₃.

p.4, Line 9: rephrase: The preparation process of RM powdery/honeycomb catalyst was illustrated in **Fig. 1**. Original RM (**Table 1a**) and 50 wt.% H₂SO₄ were mixed at a molar ratio of 1/1.2. Then the whole mixture was transferred into steel autoclave and maintained at 150 °C for 10 h, during which the alkaline/alkaline earth metal was leached, the bulk oxides such as Al₂O₃, Fe₂O₃ and TiO₂ may also partly dissolved by H₂SO₄ to form soluble aluminum sulfate, ferric sulfate and titanyl sulfate, but the SiO₂ was nearly insoluble [16]. After the treatment by H₂SO₄, the composite was washed for several times and neutralized to pH value of 8 with NH₃ aqueous to remove the alkaline/alkaline earth metal. The product of Al(OH)₃, Fe(OH)₃, Ti(OH)₂ and SiO₂ was obtained in the neutralization process.

Reference

16. Sushil S, Batra VS. 2008 Catalytic applications of red mud, an aluminium industry waste: A review. *Appl. Catal. B-Environ.* **81**, 64-77. (10.1016/j.apcatb.2007.12.002)

5. (p. 10, l. 11) The BET is not a test, as previously indicated, and this method is usually used for the calculation of specific surface area (BET surface area), excepting for microporous materials. This method does not provide information about the pore size distribution, as is indicated.

Discussion: The corresponding part has been revised as suggested.

p.9, Line 12: rephrase: The BET surface area measurement is also performed to have a deep insight into the change of surface area and pore-size distribution. **Fig. 3** illustrates the N₂ adsorption-desorption isotherms curves (**Fig. 3C**) and the corresponding pore size distribution (**Fig. 3D**) of catalysts, whose isotherms are close to type IV with a H₃-type hysteresis loop, indicating the typical mesoporous characteristics. Obviously, the pore size distribution of fresh RM catalyst (**Fig. 3C-b**) is broader than original RM (**Fig. 3C-a**). The BET surface area, pore volumes, average pore diameter are listed in **Table 2**.

6. It would be very useful to study catalysts by X-ray photoelectron spectroscopy (XPS) in order to get insights into the surface composition of fresh and used RM catalysts.

Discussion: The main focus of this paper is providing an available method to the disposal of RM solid waste as deNO_x catalyst. And the RM based catalyst showed good activity and stability as demonstrated in Fig. 5 and Fig. 6. Thus, the difference in surface composition for the fresh and used RM catalysts was not discussed and the XPS was not performed in the manuscript.

7. The scale of the x-axis of Figure 3D does not allow to distinguish clearly the evolution of pore size distribution curves. The representation until 100 nm, or less, could be enough.

Discussion: Fig. 3D has been checked and corrected in the revised manuscript as suggested.

Fig. 3 The XRD (A), TG (B), N₂ adsorption-desorption isotherms curves (C) together with the corresponding pore size distributions (D) patterns of: (a) original RM, (b) fresh RM catalyst, (c) the used sample for 10 h, (d) the used sample for 20 h, (e) the used sample for 30 h, (f) the used sample for 40 h. Reaction condition: [NO] = [NH₃] = 650 ppm, [O₂] = 3 vol.%, [SO₂] = 600 ppm, [H₂O] = 10%, N₂ balanced.

8. Have they detected the formation of a particular metal sulfate, since the percentage of SO₃ is high. By considering the formation enthalpy of different potential metal sulfates, which is the most probable to be formed in the presence of Fe(III), Al(III) and Ti(IV)?

Discussion: This is a good suggestion. Fe₂(SO₄)₃ is the most possible metal sulfates by

considering the formation enthalpy. However, the peak of $\text{Fe}_2(\text{SO}_4)_3$ or other metal sulfates was not found after carefully compared XRD pattern with the standard card. It may exist in the form of adsorptive sulfate considering its comparatively low content.

9. What is the reason of the slight increase in NO conversion in the presence of H_2O and SO_2 at 450 °C (Figure 5A)?

Discussion: As shown in **Fig. 5**, the optimal temperature window of our RM catalyst is 350-400 °C in the absence of SO_2 and H_2O . Due to the oxidation of NH_3 at high temperature, NO conversion decreased with the further increased temperature. The addition of SO_2 and H_2O inhibited the NO conversion below 350 °C. However, as the temperature increased, the inhibition effect of H_2O and SO_2 decreased. Particularly, H_2O could inhibit the oxidation of NH_3 at high temperature, which is benefit to the improvement of SCR activity above 400 °C. The observed effect of H_2O and SO_2 on the de NO_x activity of RM catalysts is well consistent with these previous reports [17], and the RM catalyst may also have good stability and resistance to H_2O and SO_2 above 350 °C.

Fig. 5 The NO conversion (A) and N₂O yield (B) with/without H_2O and SO_2 over powdery catalysts. Reaction condition: $[\text{NO}] = [\text{NH}_3] = 650$ ppm, $[\text{O}_2] = 3$ vol.%, $[\text{SO}_2] = 600$ ppm, $[\text{H}_2\text{O}] = 10$ vol.%, N_2 balanced.

p.13, Line 14: rephrase: Besides, H_2O could inhibit the oxidation of NH_3 at high temperature, which is benefit to the improvement of SCR activity above 400 °C. The observed effect of H_2O and SO_2 on the de NO_x activity of RM catalysts is well consistent with these previous reports [17], and the RM catalyst may also have good stability and resistance to H_2O and SO_2 above 350 °C.

Reference

17. Li CM, Zeng H, Liu PL, Yu J, Guo F, Xu GW, Zhang ZG. 2017 The recycle of red mud as excellent SCR catalyst for removal of NO_x . *Rsc Adv.* **7**, 53622-53630. (10.1039/c7ra10348d)

10. The total acidity can be evaluated from NH_3 -TPD data. An explanation, taking into account the chemical composition of original RM and fresh RM catalyst, must be given to justify the drastic difference between the corresponding desorption curves (Figure 7D). The existence of strong Brönsted acid sites and Lewis acid sites is put forward by the authors, but nothing is said

about the nature of chemical species responsible of these acid sites.

Discussion: As shown in **Table 1**, original RM mainly consists of Fe₂O₃, Al₂O₃, SiO₂, TiO₂, Na₂O, CaO and K₂O etc. The high Na₂O content and other alkaline-earth metals in the original RM will interact with the major active components during catalytic reaction, resulting in the decrease of surface area and activity. After the acid-hydrothermal and neutralization method, the majority of alkali/alkaline-earth metals were removed, which may increase the active acid sites as well as the NH₃ adsorption capacity [17]. As for the used RM catalyst, the residual sulfate after SCR reaction in the presence of SO₂/H₂O may further increase the NH₃ adsorption. NH₃ was mainly absorbed on the Brønsted acid sites at relative low temperature to form NH₄⁺, and then the reaction between NH₄⁺ and NO occurred to generate N₂ and H₂O. While the coordinated NH₃ bound to Lewis acid sites usually response for the SCR activity at high temperature. Both of the Brønsted acid sites and Lewis acid sites are important in the SCR reaction with NH₃.

p.17, Line 12: rephrase: The oxidizability of NH₃ is related to the acid sites of RM catalysts, thus NH₃-TPD test was carried out as profiled in **Fig. 7D**. It can be seen that the adsorption capacity of NH₃ increased significantly for the fresh RM sample compared with the original one due to the elimination of alkali/alkaline-earth metals. The desorption peak centered at about 100 °C can be assigned to physically adsorbed NH₃. Another obvious desorption peak can be observed from 300 °C to 500 °C, which may be due to the desorption of ionic NH₄⁺ bound to strong Brønsted acid sites and coordinated NH₃ bound to Lewis acid sites. It was reported [17, 27] that NH₃ was mainly absorbed on the Brønsted acid sites at relative low temperature to form NH₄⁺ and then react with NO to generate N₂ and H₂O. While the coordinated NH₃ bound to Lewis acid sites usually response for the SCR activity at high temperature. Both of the Brønsted acid sites and Lewis acid sites are important in the SCR/SCO reaction with NH₃. As for the used RM catalyst, the residual sulfate may further increase the NH₃ adsorption. These strong acid sites between 300 °C and 500 °C will facilitate the strong adsorption of ammonia and will be also beneficial for the NH₃-SCO reaction, resulting in the decreased SCR selectivity. All of these results indicate that the severe oxidation of NH₃ resulted in the low actual ratio of NH₃/NO and thus low reaction efficiency of SCR.

Reference

17. Li CM, Zeng H, Liu PL, Yu J, Guo F, Xu GW, Zhang ZG. 2017 The recycle of red mud as excellent SCR catalyst for removal of NO_x. *Rsc Adv.* **7**, 53622-53630. (10.1039/c7ra10348d)
27. Wang T, Wan ZT, Yang XC, Zhang XY, Niu XX, Sun BM. 2018 Promotional effect of iron modification on the catalytic properties of Mn-Fe/ZSM-5 catalysts in the Fast SCR reaction. *Fuel Process. Technol.* **169**, 112-121. (10.1016/j.fuproc.2017.09.029)

11. It is necessary to include in situ DRIFTS experiments of reactants in order to elucidate the mechanisms involved in SCR activity and ammoxidation. In this sense, is it actually an ammoxidation process leading to nitriles by reaction of ammonia and oxygen, using alkenes as substrate?. If this is not the case, the nature of oxidation products must be explained.

Discussion: The *in situ* DRIFTS study of RM based catalyst have been done in our previous work, which showed that the NH₃-SCR reaction over RM catalyst follows the Eley–Rideal mechanism at high temperature [17]. Besides, the detailed mechanism of NH₃ oxidation was well elaborated by other reports for Fe based catalyst[8, 23], and the main purpose of this manuscript is to find new way to lower down the ammoxidation rate at high temperature.

Reference

8. Fabrizioli P, Burgi T, Baiker A. 2002 Environmental catalysis on iron oxide-silica aerogels: Selective oxidation of NH₃ and reduction of NO by NH₃. *J. Catal.* **206**, 143-154. (10.1006/jcat.2001.3475)

17. Li CM, Zeng H, Liu PL, Yu J, Guo F, Xu GW, Zhang ZG. 2017 The recycle of red mud as excellent SCR catalyst for removal of NO_x. *Rsc Adv.* **7**, 53622-53630. (10.1039/c7ra10348d)

23. Long RQ, Yang RT. 2002 Selective catalytic oxidation of ammonia to nitrogen over Fe₂O₃-TiO₂ prepared with a sol-gel method. *J. Catal.* **207**, 158-165. (10.1006/jcat.2002.3545)

12. A paragraph comparing the present results with data already reported in the literature should be very useful for readers.

Discussion: This question is the same to **Issue 8**. Please consult the above response.

13. Minor errors: (Figure 1) Molding additive?, Grind and sieve instead of Grind and seive; (p. 7, l. 22) N₂O formation, not information.

Discussion: The errors have been checked and corrected in the revised manuscript as suggested.

Fig. 1 Schematic diagram for preparation of RM catalyst/honeycomb.

p.7, Line 5: rephrase: The NO conversion, N₂O yield and NH₃ conversion were calculated according to the following equations:

$$NO \text{ Conversion} = \frac{[NO]_{in} - [NO]_{out}}{[NO]_{in}} * 100\% \quad (1)$$

$$N_2O \text{ yield} = [N_2O]_{out} - [N_2O]_{in} \quad (2)$$

$$NH_3 \text{ Conversion} = \frac{[NH_3]_{in} - [NH_3]_{out}}{[NH_3]_{in}} * 100\% \quad (3)$$

Where [NO]_{in} and [NO]_{out} represent the concentration of gaseous NO in the inlet and outlet, respectively.